# Enhanced Vasculogenic Capacity Induced by 5-Fluorouracil Chemoresistance in a Gastric Cancer Cell Line

**DOI:** 10.3390/ijms22147698

**Published:** 2021-07-19

**Authors:** Sara Peri, Alessio Biagioni, Giampaolo Versienti, Elena Andreucci, Fabio Staderini, Giuseppe Barbato, Lisa Giovannelli, Francesco Coratti, Nicola Schiavone, Fabio Cianchi, Laura Papucci, Lucia Magnelli

**Affiliations:** 1Department of Experimental and Clinical Medicine, University of Florence, Largo Brambilla, 3-50134 Firenze, Italy; sara.peri@unifi.it (S.P.); fabio.staderini@unifi.it (F.S.); gpp.barbato@gmail.com (G.B.); corattif@gmail.com (F.C.); 2Department of Experimental and Clinical Biomedical Sciences “Mario Serio”, University of Florence, Viale G.B. Morgagni, 50-50134 Firenze, Italy; alessio.biagioni@unifi.it (A.B.); giampaolo.versienti@unifi.it (G.V.); e.andreucci@unifi.it (E.A.); laura.papucci@unifi.it (L.P.); lucia.magnelli@unifi.it (L.M.); 3Department of Neuroscience, Psychology, Drug Research and Children’s Health, University of Florence, Viale Pieraccini, 6-50139 Firenze, Italy; lisa.giovannelli@unifi.it

**Keywords:** gastric cancer, chemoresistance, tumor angiogenesis, vasculogenic mimicry, epithelial-to-endothelial transition

## Abstract

Chemotherapy is still widely used as a coadjutant in gastric cancer when surgery is not possible or in presence of metastasis. During tumor evolution, gatekeeper mutations provide a selective growth advantage to a subpopulation of cancer cells that become resistant to chemotherapy. When this phenomenon happens, patients experience tumor recurrence and treatment failure. Even if many chemoresistance mechanisms are known, such as expression of ATP-binding cassette (ABC) transporters, aldehyde dehydrogenase (ALDH1) activity and activation of peculiar intracellular signaling pathways, a common and universal marker for chemoresistant cancer cells has not been identified yet. In this study we subjected the gastric cancer cell line AGS to chronic exposure of 5-fluorouracil, cisplatin or paclitaxel, thus selecting cell subpopulations showing resistance to the different drugs. Such cells showed biological changes; among them, we observed that the acquired chemoresistance to 5-fluorouracil induced an endothelial-like phenotype and increased the capacity to form vessel-like structures. We identified the upregulation of thymidine phosphorylase (TYMP), which is one of the most commonly reported mutated genes leading to 5-fluorouracil resistance, as the cause of such enhanced vasculogenic ability.

## 1. Introduction

Gastric Cancer (GC) is to date the sixth malignancy for incidence (5.6%) and the fourth for mortality (7.7%) globally, behind lung (11.4% incidence; 18.0% mortality), breast (11.7%; 6.9%), prostate (7.3%; 3.8%), colorectal (10.0%; 9.4%), liver (4.7%; 8.3%) carcinomas and non-melanoma skin cancers (6.2%; 0.6%) [1]. Even if earlier diagnosis and better therapies have improved the Overall Survival (OS), GC is often diagnosed at an advanced stage, when the tumor is inoperable and only chemotherapy may be a useful approach [2]. Common chemotherapeutic agents, consisting of Fluoropyrimidine and Platinum-based ones, are still widely used worldwide, while Irinotecan, a topoisomerase I inhibitor, and Taxanes are more recent therapies that have shown reduced toxicity [3]. Tumor vascularization plays a fundamental role in cancer progression and metastasis, allowing tumor cells to get continuous access to oxygen and nutrients, evade the host immunosurveillance and reduce the intake of chemotherapeutic agents [4]. Indeed, during tumor growth, cancer cells release several angiogenic factors such as the Vascular Endothelial Growth Factor (VEGF) and the Fibroblast Growth Factor-2 (FGF-2) that dysregulate the homeostatic tissues equilibrium, causing the degradation of the endothelial basement membrane due to the activation of matrix metalloproteinases (MMP) [5]. The increased vascular permeability and the high concentration of pro-angiogenic factors lead to the sprouting and elongation of tissue endothelial cells to form new vessels (angiogenesis) as well as to the recruitment of circulating endothelial cells (vasculogenesis) [6]. However, the uncontrolled release of angiogenic factors and cytokines often results in the creation of leaky vessels, typical of the tumor architecture, which are easily exploited by cancer cells to intravasate into the blood and lymphatic stream and extravasate to form metastases in distant sites [7]. Angiogenesis and vasculogenic mimicry (VM) are the main processes that contribute to tumor vascularization, generating an intricate net of vessels composed of a mosaic of endothelial and tumor cells. While angiogenesis is commonly described as a complex mechanism including the remodeling of extracellular matrix, the formation of the Tip-Stalk cell hierarchy and the involvement of pericytes, VM is typically characterized by highly perfused blood lacunae not surrounded by endothelium but composed of tumor cells with significant deposition of matrix proteins and it is often associated with highly invasive and metastatic tumors, frequently paired with a poor prognosis [8]. VM, which was firstly described by Maniotis and colleagues in 1999 [9], is exploited by high-plasticity cancer cells or by the induction of the Epithelial-to-Mesenchymal Transition, triggered by hostile microenvironment conditions such as hypoxia and acidity [10]. There are other two processes described for the formation of new blood vessels: the segmentation of a pre-existing blood vessel through the involvement of the interstitial columnar structures, commonly referred to as intussusceptive angiogenesis [11] and the vessel co-option, frequently observed in many malignancies, which is the migration of tumor cells along the walls of a blood vessel to generate a new complex structure [12]. In previous years, PAS-CD31 was deemed to be the golden standard to distinguish between angiogenesis and VM. More recently, an alternative mechanism of neovascularization was observed, the Epithelial-to-Endothelial Transition (EET), not completely characterized yet [13]. EET is the acquisition of endothelial markers, such as CD31, VE-Cadherin, Ephrin A2 and others, by epithelial cancer cells [14]. It has been clarified that EET is induced by several microenvironmental factors, in cancer cells with high plasticity. In our study, we evaluated the effects at the cellular and molecular level of the chronic exposure of the AGS cell line to 5-fluorouracil (5-FU), cisplatin (CIS) and paclitaxel (TAX), three chemotherapeutic agents commonly used in GC treatment. While many mechanisms of chemoresistance and related mutations are well described for Asian GC patients [15], little is known about the mechanisms operating in Caucasian individuals, where GC has different features and prognosis. We aimed at the investigation of mechanisms involved in the chemoresistance of AGS cells, focusing in particular on the capacity of cancer cells to form a new vascular network. 5FU-resistant cells, which demonstrated, among the analyzed phenotypes, a significant upregulation of the thymidine phosphorylase gene (TYMP), were capable to form complex vessels via the expression of typical angiogenesis markers.

## 2. Results

### 2.1. Establishment of AGS Resistant Cell Lines

We subjected AGS cell line to chronic exposure of the three more common chemotherapeutic agents for the treatment of advanced GC (5-FU, CIS and TAX), as reported above. Briefly, low-density AGS cells were initially treated for 24 h with 5-FU (3 μM) or CIS (2 μM) or TAX (1 nM) and then plates were rinsed and cells were allowed to grow in drug-free fresh complete media until reaching sub-confluence, replenishing media every 2–3 days. Cells were then plated in new Petri dishes at low density. We regularly checked the IC_50_ and after 9 months of culture we evaluated the acquired fold-resistance by the following equation:Fold-Resistance = (IC_50_ of Resistant Cell Line)/(IC_50_ of Parental Cell Line).(1)

As shown in Figure 1, AGS subjected to 5FU demonstrated a resistance factor of more than 200 (4.4 to >1000 µM), Cisplatin of about 4.5 (15.3 to 75.1 µM) and Taxol more than 90 (1.1 to 104.2 nM). Regarding 5FUr, we were not able to calculate accurately the IC_50_ since even at the highest doses the lethality did not reach 50%. We also could not further increase the drug concentration due to the solubility limit of the selecting agent, one of the main limiting factors in how much drug can be applied to cells for an in vitro test. Moreover, doses that are close to the limit of solubility will not be taken under consideration as clinically relevant.

### 2.2. Characterization of Chemoresistant Cell Lines

Notably, when we treated the three resistant cell lines with the drugs they were not selected for, at individual concentrations, we did not observe cross-resistance (Figure 1), demonstrating that the gained, underlying resistance mechanisms, are different for each chemotherapeutic agent. Such phenomenon was further confirmed by the absence of the Multidrug Resistance Protein1-MDR1 (alias CD243 or Pgp) which was slightly detectable only in TAXr cells (Appendix A). AGS resistant cells demonstrated different sizes and morphology with respect to the wild-type ones as shown in Figure 2a. In particular, 5FUr showed some very large cellular bodies with big nuclei which might be generated by the fusion of several cells in a sort of syncytial formation, while CISr demonstrated reduced dimensions. We monitored cell proliferation through CFSE labeling assay, showing that at 24 h only CISr have a lower proliferation ratio, but after 48 h we observed a decreased proliferation index (the average number of divisions that all responding cells have undergone since the initiation of the culture [16]) in all resistant cells (Figure 2b). As shown in Figure 2c, such inhibited proliferation rate is consistent with an accumulation of 5FUr and TAXr in S phase while CISr encountered a delay in G0/G1 phase.

### 2.3. 5FUr Cells Enhanced Vasculogenic Capability

As one of the main phenomena associated with the gaining of chemoresistance is the enhanced tumor vascularization, we decided to evaluate the chemoresistant cells vasculogenic potential plating AGS WT and resistant cells on Matrigel. While we observed a moderate tube formation in the AGS WT and 5FUr samples during the first 6 h (Data not shown) after 24 h only 5FUr cells were able to keep the tubular organization (Figure 3a). As reported in the histogram below we analyzed three of the most important angiogenesis parameters (definition provided in Section 2.6) evaluating that only 5FUr enhanced significantly the tubular formation. Such a vascularization is, as expected, incomplete and aberrant but, as already reported, GC is often capable to perform vascular-like structures to reach new sources of oxygen and nutrients [17]. We then evaluated several endothelial or VM-related markers through WB (Figure 3b) and we demonstrated an upregulation of VEGFR2 and Ephrin A2 in 5FUr only, while Galectin-3 was upregulated in both 5FUr and CISr. Laminin ɣ2-chain was found to be upregulated in all the resistant cell lines. Even though several VM markers are expressed on the three chemoresistant phenotypes, the higher expression of EphA2 may play a fundamental role in vascular formation and stabilization [17].

### 2.4. Evaluation of the Cooperation with Endothelial Colony Forming Cells

To better understand if the endothelial-like phenotype observed in the 5FUr cells is associated with a proper ability to collaborate with pre-existing endothelial cells, forming in such a way a complex mosaic structure, AGS resistant cells were plated alone or co-cultured with endothelial colony-forming cells (ECFCs), which are a subpopulation of endothelial progenitor cells (EPCs), capable of proliferating and forming tubular vessel-like structures in vitro. As shown in Figure 4, AGS WT and 5FUr were able to perform tubulogenesis when plated on Matrigel, while CISr and TAXr organized themselves in clusters incapable to form vessels. The number of junctions evaluated was significantly higher for 5FUr and the formed vessels survived more than any other phenotype. When co-cultured with ECFCs, all cell lines, except TAXr, demonstrated to join angiogenesis, but while CISr and AGS WT participated passively by coating the ECFCs-derived vessels in a sort of vessel co-option, 5FUr formed their network perfectly anastomosing with the one produced by ECFCs and stabilizing vessels with a greater caliber. As internal control, ECFCs alone were reported in Appendix A.

### 2.5. TYMP as the Key between Vascularization and 5-FU Metabolism

It is to date well established the involvement of the thymidine phosphorylase-thymidylate synthase axis in 5-FU metabolism and activation. As already reported [18] TYMP in particular plays a major role also in angiogenesis interacting with several pro-angiogenic factors. Indeed, evaluating such markers by qPCR, we evidenced a significant upregulation of both TYMP and TYMS in 5FUr cells, as shown in Figure 5a. We also evaluated if the previous upregulation of endothelial-related markers might be led by TYMP or TYMS through a bioinformatic analysis via STRING tool. We identified, as shown in Figure 5b, that TYMP is closely associated with VEGFR2 (KDR) and obviously with TYMS while it does not interact directly with EphA2 (EPHA2) and Galectin-3 (LGALS3). Such interaction is instead mediated by KDR. Laminin γ2 (LAMC2) was found completely unrelated to TYMP and TYMS which might explain the overexpression in all the chemoresistant cells (Coexpression patterns are available in Appendix A). We also found it interesting that a typical endothelial marker such as CD31 (PECAM-1), which is commonly associated with VEGFR2 and Galectin-3 [19], is also closely associated with TYMP. Being VM and Epithelial to Endothelial-Transition (EET) two similar phenomena, we decided to evaluate the expression of CD31 which is a specific marker of endothelial cells and thus is not expressed during VM. Therefore, we analyzed CD31 in AGS WT and 5FUr, being the only two phenotypes among the four analyzed demonstrating a significant capability to form vessels. As shown in Figure 5c, CD31 is unexpectedly expressed in 5FUr cells, while it is completely absent in AGS WT, determining the typical trait of the EET. To better understand if such a phenotype might be caused by the thymidine phosphorylase, we silenced TYMP expression (Appendix A). As expected, we observed a reduced proliferation ratio in both AGS and 5FUr silenced cells with respect to the control ones (Figure 5d), having TYMP the physiological function to fuel the pool of nucleotides, catalyzing the phosphorylation of thymidine or deoxyuridine to thymine or uracil [20]. Evaluating the vascular formation, we demonstrated that TYMP expression is fundamental for 5FUr as its downregulation led to a diminished vessel generation in terms of nodes, junction and segments while it did not interfere with AGS WT (Figure 5e).

### 2.6. Thalidomide Partially Reverts 5-FU Resistance

To normalize tumor angiogenesis induced by 5-FU treatment, we evaluated Rapamycin, Genistein and Thalidomide as anti-VM and anti-angiogenic drugs [21,22,23] on AGS WT and 5FUr, the only cell phenotype that demonstrated to have a significant vasculogenic capability. While Rapamycin and Genistein showed significant inhibition of cell viability (Appendix A), Thalidomide did not affect AGS WT or 5FUr if not in combination with 5-FU.

Indeed, as shown in Figure 6a, 5FUr cells were sensitive to Thalidomide when combined with 5-FU, demonstrating a partial recovery of chemosensitivity to this drug. Thalidomide was also capable to inhibit the invasive ability of both AGS WT and 5FUr albeit with higher efficiency in the 5-FU resistant cells, as shown in Figure 6b. It is important to notice that 5FUr demonstrated reduced invasive ability with respect to their wild-type counterpart which might be partially explained by their principal need to resist the hostile environment generated by the chemotherapeutic agent, consuming higher resources otherwise required to actively invade the surrounding tissues. Furthermore, Thalidomide inhibited 5FUr tubulogenic activity as well, as shown in Figure 6c. Finally, we determined that Thalidomide was capable to reduced 5FUr cells migration ability while it did not exert any significant effect on AGS Wt (Figure 6d).

## 3. Discussion

Treating AGS cells chronically with sub-lethal doses of 5-FU, Cisplatin or Paclitaxel, respectively, we were able to select a cell subset exhibiting drug-resistant phenotype, reduced proliferation index and different morphology with respect to the time-zero wild-type cells. We speculated that several molecular pathways might be involved in slowing down cell proliferation, which may well represent a passive mechanism enabling cancer cells to avoid the effects of chemotherapy. Moreover, chemoresistant cells might have a higher energy demand with respect to sensitive ones in order to resist the sub-lethal doses of chemotherapeutic agents and therefore they are forced to slow down their proliferative capability. This phenomenon is routinely observed in clinical practice, when repeated chemotherapy treatments cause an initial reduction of the tumor mass, until a resistant cell subset arises from the heterogeneous pool of mutated cells and starts to grow back. These chemo- and radio-resistant cells often gain invasive and migratory capacities leading to cancer cell spreading and formation of metastases in distant tissues [18]. The tumor microenvironment might also contribute to chemoresistance, generating an intricate net of endothelial vessels to allow the tumor mass to reach a continuous source of nutrients and oxygen but preventing the delivery of chemotherapy. To date, several angiogenesis mechanisms have been identified in normal tissues and tumors: in mammalian embryos, angioblasts differentiate into endothelial cells, a process known as vasculogenesis, while later endothelial cells expand the vascular network throughout the so-called angiogenesis. Once the early vascular network is completed, vessels can split by intussusception, enriching the network. Beyond these typically physiological processes, tumor cells may also create a new vascularization by co-opting the pre-existing vasculature or lining and supporting endothelial cells, in the so-called vasculogenic mimicry (VM) [20]. So, we do believe that cancer cells may contribute to the tumor vasculature depending on their potential: more differentiated tumor cells can generate vessels through VM, while stem-like cells might form vasculature networks through differentiation into endothelial-like cells. Indeed, tumor cells which express markers associated with the stem cell functional phenotype, such as Notch, Wnt, ABCB5, CD133, CD166, nestin, and c-kit, and significantly contribute to the establishment of tumor vasculature have been observed in aggressive melanoma and glioblastoma [24,25]. Our cell model demonstrated very different vasculogenic capabilities, as CISr and TAXr cells completely lost their capacity to organize vessels with respect to the moderate proficiency of WT AGS cells. On the contrary, 5FUr cells showed a significantly improved ability to generate vasculature, albeit obviously incomplete and disorganized. Chronic exposure of AGS cells to 5-FU (5FUr cell generation) induced indeed typical endothelial markers which were absent (CD31) or low-expressed (VEGFR2 and EphA2) in the other chemoresistant cell phenotypes. In particular, EphA2 had been already reported to be overexpressed in GC with a high grade of VM [17,26]. Moreover, while TAXr cells were completely incapable to form a vascular network and CISr cells can only passively connect with newly formed vessels made by endothelial cells, 5FUr cells exhibited a strong ability to actively cooperate with ECFCs to form vessels. The REGARD (2009–2012) [21] and the RAINBOW (2010–2012) [22] clinical trials demonstrated for the first time the usefulness to combine common chemotherapies with an anti-angiogenic treatment for advanced GC. Indeed, thanks to the results of these two studies, ramucirumab was approved by FDA in 2014 as second-line treatment of advanced GC. Moreover, it was already reported that the use of enalapril is capable to restore the 5-FU sensitivity of colon cancer cells and significantly suppress tumor growth, metastasis and angiogenesis [27]. Tumor angiogenesis was also found increased in sorafenib-resistant hepatocellular carcinomas through the upregulation of VEGF signaling associated with cancer stem cells [28]. TYMP is commonly reported to play a major role in 5-FU chemoresistance and is often found to be overexpressed in both diffuse and intestinal gastric adenocarcinomas [18]. TYMP is also closely associated with the metastatic phenotype, although no statistically relevant data about its expression in chemoresistant patients is available from the databanks [29]. In this context, TYMP might be the link between the two phenomena, chemoresistance and increased vasculogenesis, as CISr and TAXr demonstrated to be incapable to form vessels *in vitro*. In the last part of our study, we attempted to use a pharmaceutical approach to treat chemoresistance and neovascularization by subjecting 5FUr cells to the treatment with Thalidomide, which is not only an anti-angiogenic drug but also a VM inhibitor. This combination treatment resulted in a partial recovery of 5-FU sensitivity by 5FUr cells, as it determined the reduction of migration, invasion and vessel formation. A similar attempt was recently made by Leuci V. et al. treating xenografts of cells from a liver metastasis of colorectal cancer with lenalidomide, a derivative of thalidomide [30] currently approved for treating multiple myeloma and 5q- myelodysplastic syndrome [31]. However, even if promising results have been obtained, the still incomplete clarification of the action mechanism of thalidomide and its derivatives pose a great limit to their clinical use. We speculated that their action might be exerted through TYMP as a target of phthalimide/homophthalimide skeleton drugs such as thalidomide [32]. Indeed, Kita T. et al. reported that N(α)-phthalimide glutarimide derivatives are able to selectively target the platelet-derived endothelial cell growth factor (PD–ECGF) which is structurally identical to TYMP. If this mechanism of action is shared by thalidomide, TYMP inhibition could revert TYMP-induced chemoresistance and the relative increased vascular formation. It is clear that a complex phenomenon such as chemoresistance cannot most probably be explained just with the upregulation of a single enzyme and that other players are obviously involved in the modulation of the TYMP/TYMS axis [18].

## 4. Materials and Methods

### 4.1. Cell Lines and Chemotherapeutic Agents

AGS gastric cancer cells were purchased from ATCC (CRL-1739). Cells were maintained in F12K medium (Corning, Milano, Italy) supplemented with 10% FBS (Euroclone, Milano, Italy) and 1% L-glutamine (Euroclone). Endothelial Colony Forming Cells (EFCFs) (kindly gifted by Prof. Calorini’s Laboratory) were isolated from human umbilical cord blood of healthy newborns as previously described [33] and were maintained in Endothelial Basal Medium (EBM-2, Lonza distributed by Euroclone). Cells were tested every two weeks for Mycoplasma by PCR using two universal primers (MGSO and GPO1) [34]. 5-fluorouracil (F6627), cisplatin (P4394) and paclitaxel (T7402) were purchased from Sigma-Aldrich and stock solutions were prepared according to the manufacturer’s instructions. In order to obtain 5-FU-resistant (5FUr), CIS-resistant (CISr) and TAX-resistant (TAXr) cell lines, low-density AGS cells were initially treated for 24 h with 5-FU (3 μM), CIS (2 μM) or TAX (1 nM), and then plates were rinsed and cells were grown in drug-free fresh complete media until reaching sub-confluence, replenishing media every 2–3 days. Cells were then plated in new Petri dishes at low density. This procedure was repeated for about 9 months. Similar methods were already reported in the literature by McDermott et al. [35]. From this time on, cells were constantly maintained in presence of chemotherapeutic agents at these concentrations.

### 4.2. IC50 Assay

Cells were plated at a density of 5 × 10^3^ per well in a 96 multiwell plate. After 24 h the culture medium was replaced and cells were treated with scalar dilutions of drugs. After 72 h cells were incubated for 2 h at 37 °C in dark with 0.5 mg/mL 3-(4,5-dimethylthiazol-2-yl)-2,5-diphenyl tetrazolium bromide (MTT) in medium without phenol red. MTT was removed and cells were lysed in 100 µL DMSO. Absorbance values were recorded at 595 nm with an automatic plate reader (Bio-Rad, Milano, Italy). All data were expressed as mean ± SD in percentage with respect to drug-untreated controls. IC50 values were calculated using Graphpad Prism software.

### 4.3. Western Blot Analysis

Aliquots of 50–80 μg of whole-cell lysates, obtained as previously described [36], were subjected to Western blotting. The primary antibodies were: galectin-3 (sc-32790), laminin γ-2 (sc-28330) and β-actin (sc-47778) from Santa Cruz Biotechnology; VEGFR2 (441053G) from Invitrogen and EphA2 (05-480) from Merck Millipore. Membranes were incubated in Odyssey Blocking Solution (Millipore, Milano, Italy) for 1 h at room temperature. Membranes were then incubated overnight at 4 °C with the primary antibody (all primary antibodies were used diluted 1:1000 in a mix of 1:1 Odyssey Blocking Solution and PBS-Tween 0.1%), washed with PBS-Tween 0.1% solution, and probed with the secondary IRDye antibody according to the manufacturer’s instructions (secondary antibody diluted 1:20,000). The protein bands were analyzed by the Odyssey Infrared Imaging System (LI-COR Bioscience, Nebraska, USA) using the Odyssey software for protein quantification.

### 4.4. Cell Cycle Analysis

Cultured cells were collected, washed with PBS, resuspended with 0.1% TritonX-100 and 0.1% (*w*/*v*) Trisodium, and stained with propidium iodide (PI, final concentration 50 µg/mL). Stained cells were subjected to the BD FACS Canto II flow cytometer (BD Biosciences distributed by DBA Italia, Milano, Italy) and analyzed with the FlowJo software (BD Biosciences).

### 4.5. Flow Cytometry Analysis

Cells were harvested with Accutase (Euroclone), washed once with cold PBS and then stained with fluorochrome-conjugated mAbs anti-CD31 (Miltenyi Biotec, Bologna, Italy) or anti-CD243 (eBioscience, distributed by Thermo Fisher Scientific, Monza, Italy) for 1 h on ice in the dark. After washing in PBS plus Bovine Serum Albumin 0.5% (BSA), cells were analyzed by flow cytometry BD FACS Canto II with Diva Software (BD Biosciences).

### 4.6. Tubulogenesis Formation Assay

µ-Plate Angiogenesis 96 well (Ibidi, Gräfelfing, Germany) were coated with 10 µL/well of Matrigel (DBA) and incubated for 1 h at 37 °C to let the matrix polymerize. 1.8 × 10^4^ cells per well were seeded and then incubated at 37 °C in a 5% CO_2_ humidified atmosphere. For the vasculogenic inhibition experiments, cells were let form vascular structures for 3 h and then Thalidomide 200 µM (Sigma Aldrich, Milano, Italy) was added. For co-culture tube formation assay, AGS WT or chemoresistant cells and ECFCs were marked following manufacturer instructions: briefly, 1 × 10^6^ cells were resuspended in 1 mL of PBS with 1 μL of CFSE or Far-Red Cell Trace stock solution (C34570 and C34572 Thermo Fisher Scientific, Monza, Italy) and incubated at 37 °C for 20 min. The cells were then seeded again in P100 plates. The day after cells were harvested and plated on glass-bottom 8 wells supports (Ibidi), pre-coated with 50 µL of Matrigel, in a ratio of 3:2 cancer cells:ECFCs for a total of 2.5 × 10^4^ cells. Images were taken using the TE300 Nikon Eclipse for fluorescence analysis while the Bio-Rad MRC 1024 ES Confocal Laser Scanning Microscope (Bio-Rad, Hercules, CA, USA) equipped with a 15 mW Krypton/Argon laser and a Nikon Plan Apo X60-oil immersion objective was used for the confocal analysis. A single composite image was obtained by the superimposition of twenty optical sections for each sample observed. The angiogenesis analyzer tool of ImageJ software was used for the quantitation of the number of nodes, as circular dots, junctions, which corresponds to groups of fusing nodes and segments, as elements delimited by two junctions, providing the statistical analysis for each experimental condition tested [37].

### 4.7. CellTrace CFSE Proliferation Assay

Cells were harvested with Accutase (Euroclone) and stained with CellTrace CFSE (Thermo Fisher Scientific) according to the manufacturer’s instruction. Cells were harvested 24 and 48 h after the start of the experiment and compared with control (T0). Cells were then fixed and analyzed by flow cytometer through ModFitLT software (BD Biosciences).

### 4.8. Invasion Assays

Invasion ability was evaluated using transwell polycarbonate filters with 8 μm diameter pore size (Millipore) coated with 0.25 μg/μL Matrigel (BD, Corning) and seeding 1.0 × 10^5^ cells in the upper compartment. No FBS gradient was used. After 6 h of incubation, filters were fixed in methanol overnight at 4 °C and non-migrated cells were removed. Invasive cells were stained with Diff Quick solution (Microptic, Barcelona, Spain) to allow microscopy counting. Pictures were analyzed using ImageJ software.

### 4.9. Wound Repair Assay

Migration ability was assessed using 4 Well Insert (Ibidi). Briefly, 1.0 × 10^5^ cells were seeded in the wells of a culture insert located at the center of a 35 mm culture dish. After 24 h the culture insert was removed to reveal the wound gap of 500 μm. Then, cells were washed with PBS to remove floating cells and fresh medium with or without Thalidomide (200 μM) was added. Pictures were taken at regular intervals until 72 h using a 10× phase contrast objective (Nikon, Amsterdam, The Netherlands). Wound area was measured using ImageJ software via MRI wound healing plugin [38].

### 4.10. Protein-Protein Interaction Analysis

STRING v11 was used to generate and visualize a complex map of all the known and predicted interactions among the queried proteins setting 0.400 as the minimum required confidence for the interaction score [39]. The proteins involved in angiogenesis were selected from the most recent topic publications available in the literature.

### 4.11. Statistics

Results are expressed as means ± SD. Multiple comparisons were performed by the Student test or One-way or Two-way ANOVA using GraphPad Prism. Chemoresistant cell lines were compared to AGS WT unless otherwise indicated in the figure legends. Statistical significances were accepted at *p* < 0.5.

## 5. Conclusions

We observed for the first time in a Caucasian GC cell line a close relationship between the chemoresistance to 5-FU and tumor vascularization. Therefore, we speculated that chronic exposure to 5-FU might select a pool of cells with the capability to differentiate into an endothelial-like phenotype, a property typical of highly aggressive tumor cells with an enhanced metastatic potential. We found this phenomenon related only to the 5FUr cell phenotype, but not to CISr and TAXr cells. Indeed, every chemotherapeutic agent may select preferentially a subclone of the tumor population leading to a predominant phenotype, so it is of primary importance to focus research on the feasibility of combined treatments based on the selective effect of each drug on individual pathways and on the genes mutations specifically involved.

## Figures and Tables

**Figure 1 ijms-22-07698-f001:**
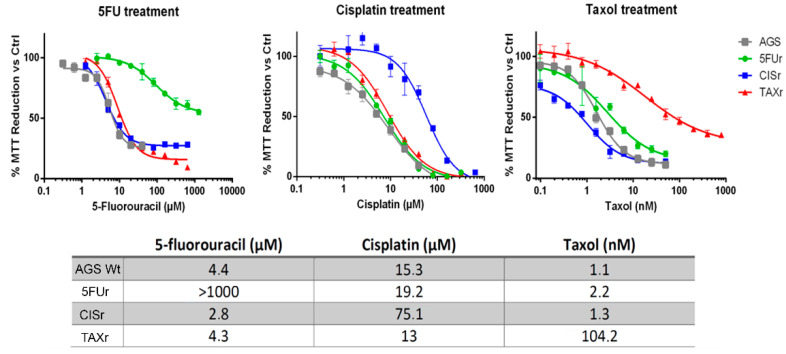
Establishment of AGS-resistant cell lines. AGS cell line chronically subjected to 5-FU, CIS or TAX, called hereafter 5FUr, CISr and TAXr respectively, showed increased IC_50_. Cells were exposed to a range of drug concentrations for 72 h and cell viability was assessed by MTT assay. Results of representative experiments are shown. Data are presented as mean ± SD (*n* = 5).

**Figure 2 ijms-22-07698-f002:**
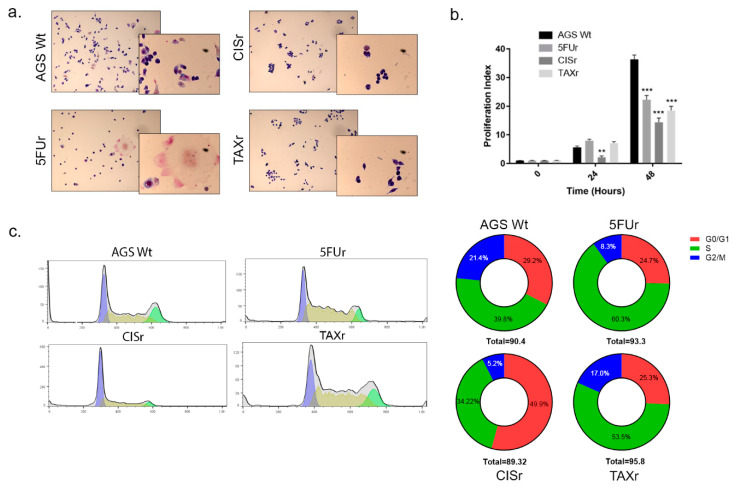
Characterization of Chemoresistant Cell Lines. (**a**) Images of AGS WT and chemoresistant cells. Cells were fixed and stained with Hematoxylin and Eosin. (**b**) Proliferation Index: fold expansion during cell culture (ratio of final cell count to starting cell count) as defined in ModFitLT. (**c**) Cell cycle profile was evaluated using FACS Canto II collecting 10,000 events per sample and analyzing using the FlowJo software. Cell cycle curves identifying the G0/G1, S and G2/M phases were plotted (left) and the area under the curves was analyzed and presented in pie charts (right). The total percentage of cells is indicative of the three phases analyzed, avoiding the sub-G0/G1 and the over-G2/M representing debris and apoptotic cells. Even if slightly different, the total percentage of events analyzed is not significant among the four cell lines. Data are presented as mean ± SD (*n* = 3) ** *p* < 0.001; *** *p* < 0.0001 (Two-way ANOVA test).

**Figure 3 ijms-22-07698-f003:**
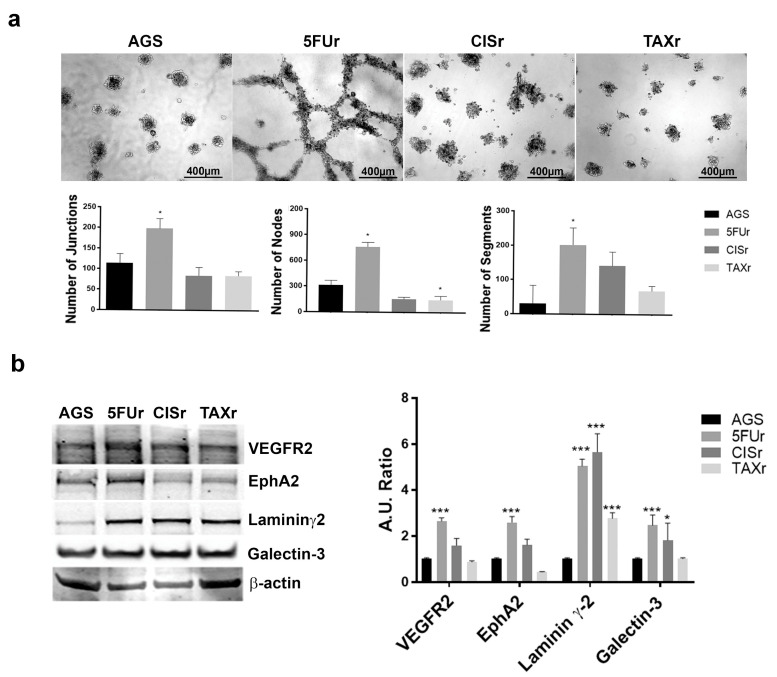
5FUr cells enhanced vasculogenic capability. (**a**) AGS WT and chemoresistant cells were plated on Matrigel and pictures were taken after 24 h. The representative images and the number of Junctions, Nodes and Segments of tubes are presented. Images were captured at 10X magnification—scale bar 400 µm. (**b**) Whole-cell lysates were analyzed by Western Blot for VM-related markers expression. β-Actin was used as a loading control. Densitometric analysis was performed using ImageJ software. Data are presented as mean ± SD (*n* = 3). * *p* < 0.01; *** *p* < 0.0001 (Two-way ANOVA test).

**Figure 4 ijms-22-07698-f004:**
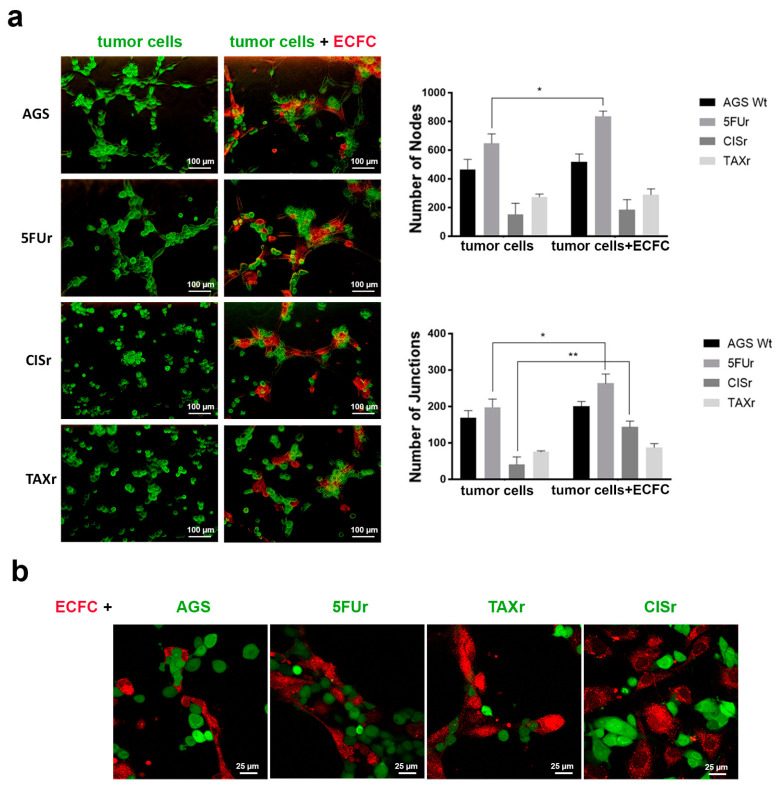
Evaluation of the cooperation with Endothelial Colony Forming Cells. AGS WT and chemoresistant cells were plated with or without ECFCs on Matrigel and pictures were taken at 6 h. The representative images and the number of Nodes and Junctions of tubes are presented. Images were captured at 10X magnification at the fluorescence microscope—scale bar 100 µm—(**a**) and 60X -oil immersion at the confocal microscope—scale bar 25 µm—(**b**). Data are presented as mean ± SD (*n* = 3). * *p* < 0.01; ** *p* < 0.001 (Two-way ANOVA test).

**Figure 5 ijms-22-07698-f005:**
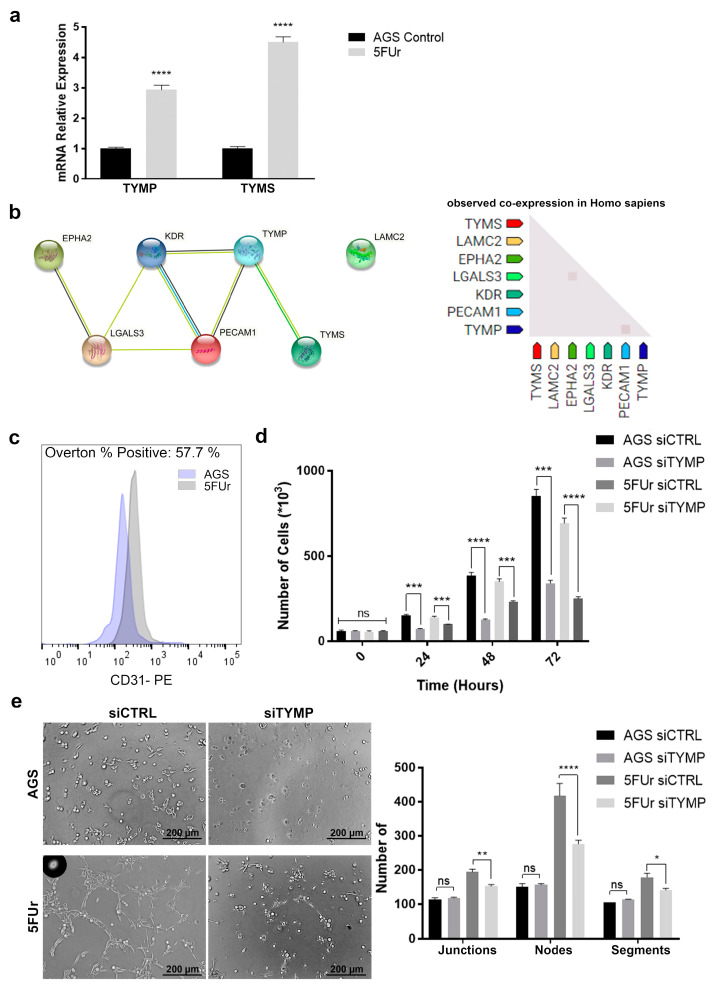
TYMP as the key between vascularization and 5-FU metabolism. (**a**) Total RNA isolated was subjected to RT–PCR analysis of TYMP and TYMS genes. B2M and GAPDH were used as a loading control (*n* = 3). (**b**) A comprehensive map of the known and predicted interactions among the previously analyzed angiogenesis-related markers and CD31. (**c**) Cells were tested for CD31 expression by FACS analysis (Overton 57.7% *p* < 0.0001) (*n* = 3). (**d**) Cellular growth counting the total number of cells 24, 48 and 72 h after siRNA transfection. (*n* = 3). (**e**) Cells were plated on Matrigel and pictures were taken after 24 h (*n* = 3)—scale bar 200 µm. The representative images and the number of Junctions, Nodes and Segments are presented. Images were captured at 20× magnification. Data are presented as mean ± SD. ns means not statistically significant; * *p* < 0.05; ** *p* < 0.01; *** *p* < 0.001; **** *p* < 0.0001 (Two-way ANOVA test).

**Figure 6 ijms-22-07698-f006:**
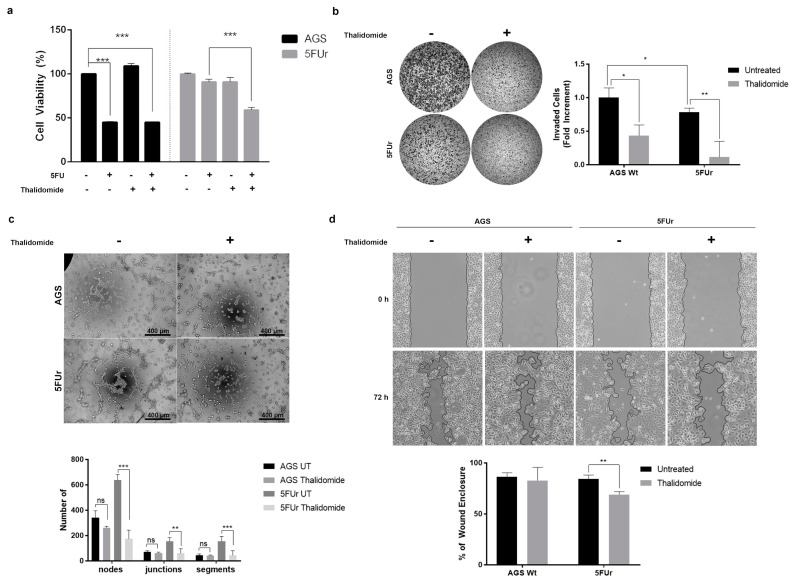
Thalidomide reverts 5-FU resistance. (**a**) Cells were exposed to Thalidomide 200 µM for 72 h and cell viability was assessed by MTT assay. (*n* = 5). (**b**) Cells were harvested and plated on Thalidomide 200 µM for 6 h (*n* = 3). (**c**) AGS WT and 5FUr were plated on Matrigel and after 3 h Thalidomide 200 µM was added. Pictures were taken at 3 h after the treatment. The representative images and the number of Nodes, Junctions and Segments are presented (*n* = 3). Images were captured at 10X magnification—scale bar 400 µm. (**d**) Representative images of wound healing after scratching (T0) and 72 h later. Quantification (below) of the wound enclosed area (*n* = 4). Results of representative experiments are shown. Data are presented as mean ± SD. ns means not statistically significant; * *p* < 0.01; ** *p* < 0.001; *** *p* < 0.0001 (Two-way ANOVA test).

## Data Availability

The data that support the findings of this study are available from the corresponding author upon reasonable request.

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
