# Peer review of "Enhanced Vasculogenic Capacity Induced by 5-Fluorouracil Chemoresistance in a Gastric Cancer Cell Line"

_ijms, 2021, doi:10.3390/ijms22147698_

Round 1

Reviewer 1 Report

The authors have investigated the role and mechanism of acquired chemo-resistance  on vascularization using gastric cancer cell line AGS. The authors have subjected AGS cell line to different chemotherapeutic agents namely 5-Fluorouracil, Cisplatin or Paclitaxel to establish chemo-resistant cell lines. The authors have shown AGS-5 Fluorouracil resistant cell line have increased vascularization. The authors have also shown AGS-5 Fluorouracil resistant cell line have increased expression of thymidine phosphorylase (TYMP) and knockdown of TYMP decreased the vasculogenic potential of AGS-5 Fluorouracil resistant cell line. It is interesting study but the manuscript could not be considered for publication in the current form for following reasons.

Major concerns:

1) The manuscript is poorly written and numerous sentences must be rephrased in the results section. Could authors please carefully review the manuscript and correct them.

2) Did authors perform MTT assay on resistant cell lines to rule-out acquired cross-resistance for non-selective chemotherapeutic agents? If so please include the data.

3) Could authors please clarify whether AGS WT proliferation rate is compared to resistant cell lines in Figure 2B? If so the authors must show statistics sign accordingly to denote which groups are being compared in the analysis. 

4) Figure 2B represent cell proliferation rate of AGS WT and resistant cell lines at different time points. It is not clear what authors meant "the average number of divisions that all responding cells have undergone since the initiation of the culture" on Ln # 179-180. The figure 2B data do not present number of divisions the cells have undergone and authors must consider their interpretation and description. 

5) Figure 3C is a snapshot on percentage of cells at different phases of cell cycle for the timepoint cells were analyzed. The authors have stated "inhibited proliferation rate is consistent with an accumulation of 5FUr and TAXr in S phase while CISr encountered a delay in G0/G1 phase." 

5A) Could authors please comment whether it is decreased or inhibited proliferation rate for respective resistant cell lines? 

5B) Did authors perform any additional experiments to attribute the proliferation rates of resistant cell lines to delay in  respective phases of cell cycle as stated in Ln # 181-183?

6) Could authors please describe methods used in the study in materials and methods section. Authors did not describe CFSE labelling assay, the time points used for cell cycle analysis.

7) Could authors please indicate percentage of cells in different phases of cell cycle for Figure 2D. The total gated population represents from 89.32% to 95.8%. Could authors please comment on status of remaining percentage of cells for respective groups?

8) Could authors please include the details on antibody dilutions used in the study and catalog numbers of respective antibodies.

Minor concerns:

1) Could authors please rewrite the sentence in Ln # 37-39. The information is very generic and do not provide clarity on ranking of cancer types by mortality and incidence rates. The authors could include respective cancer types in parenthesis next to malignancy and mortality. 

2) Please rephrase the sentence in Ln # 127-129.

3) The authors must introduce the complete name with abbreviation first before abbreviations are used. The authors have used the abbreviations 5FUr, CISr, TAXr but were not introduced. Please elaborate the abbreviations at the first instance before abbreviations are used.

Reviewer 2 Report

This article deals with the effect of 5FU chemoresistance in the transition of GC cells  to endothelial cells and vasculogenesis. The main of the work described is interesting and merit to be further investigated to understand the role of chemoresistance in angiogenesis tumors.

Minor changes:

1) Vasculogenesic mimicry is a controversial point of view in the field of tumor angiogenesis and this point is not well presented as well as discussed. Some references are missing

2) The references are not well edited. Please check

3) In the materials and medhods please explain the rational for the use of the doses as compared that the clinical datas

4) The image in Fig2a is not clear concerning 5FUr cells. Please explain it or change it.

Major changes

1) Although the results are convincing,  in vivo experiments with the different cell lines have to be investigated in order to analyse the angiogenesis in each type of tumors. In addition, the agressivity could also be tested and compared between sensitive and chemoresistant cells with or without Thalidomide

2) Western blot and the shift of ephr2A in the 5FUr line is not clear and/or explained

Round 2

Reviewer 1 Report

The authors have investigated the role and mechanism of acquired chemo-resistance on vascularization using gastric cancer cell line AGS. The authors have subjected AGS cell line to different chemotherapeutic agents namely 5-Fluorouracil, Cisplatin or Paclitaxel to establish chemo-resistant cell lines. The authors have shown AGS-5 Fluorouracil resistant cell line have increased vascularization. The authors have also shown AGS-5 Fluorouracil resistant cell line have increased expression of thymidine phosphorylase (TYMP) and knockdown of TYMP decreased the vasculogenic potential of AGS-5 Fluorouracil resistant cell line. It is interesting study and authors have addressed most of the concerns and are requested to address following concerns before the manuscript could be considered for publication.

Concerns:

1) The authors have stated the gastric cancer ranks fifth in incidence rate and fourth in mortality rate globally. If the percentages of mortality rates stated for other cancers in Ln # 38 - 41 is correct, gastric cancer would stand at second place as per the details provided by the authors in the manuscript. It is contradicting with their statement on Ln # 38. Could authors please carefully check the mortality rates and fix the issue.

2) Several sentences in the manuscript could be paraphrased to convey the message clearly. It would certainly be helpful if authors choose to use the English editing service.

Reviewer 2 Report

the authors have improved the quality of the manuscript although I still think that in vivo studies on nude mice are lacking ci=oncerning the validation of the in vitro results and the importance of VM.
